# A High Body Mass Index and the Vacuum Phenomenon Upregulate Pain-Related Molecules in Human Degenerated Intervertebral Discs

**DOI:** 10.3390/ijms23062973

**Published:** 2022-03-10

**Authors:** Masayuki Miyagi, Kentaro Uchida, Sho Inoue, Shotaro Takano, Mitsufumi Nakawaki, Ayumu Kawakubo, Hiroyuki Sekiguchi, Toshiyuki Nakazawa, Takayuki Imura, Wataru Saito, Eiki Shirasawa, Akiyoshi Kuroda, Shinsuke Ikeda, Yuji Yokozeki, Yusuke Mimura, Tsutomu Akazawa, Masashi Takaso, Gen Inoue

**Affiliations:** 1Department of Orthopedic Surgery, Kitasato University School of Medicine, 1-15-1 Minami-Ku Kitasato, Sagamihara City 252-0374, Kanagawa, Japan; kuchida@med.kitasato-u.ac.jp (K.U.); ino2510@gmail.com (S.I.); chotta0829@hotmail.com (S.T.); mikkun.nakaki.81@gmail.com (M.N.); ayumukawakubo0827@gmail.com (A.K.); nakazawa@kitasato-u.ac.jp (T.N.); tk2003@kitasato-u.ac.jp (T.I.); boatwataru0712@gmail.com (W.S.); eeiikkii922@yahoo.co.jp (E.S.); akiyoshikvroda@yahoo.co.jp (A.K.); ikedas@kitasato-u.ac.jp (S.I.); yuji0328yoko@yahoo.co.jp (Y.Y.); msm.men.36@gmail.com (Y.M.); mtakaso@kitasato-u.ac.jp (M.T.); ginoue@kitasato-u.ac.jp (G.I.); 2Shonan University of Medical Sciences Research Institute, Nishikubo 500, Chigasaki City 253-0083, Kanagawa, Japan; guccyon_s@yahoo.co.jp; 3Department of Orthopedic Surgery, St. Marianna University School of Medicine, 2-16-1 Sugao, Kawasaki City 216-8511, Kanagawa, Japan; cds00350@par.odn.ne.jp

**Keywords:** discogenic low back pain, calcitonin gene-related peptide, microsomal prostaglandin E synthase-1

## Abstract

Animal studies suggest that pain-related-molecule upregulation in degenerated intervertebral discs (IVDs) potentially leads to low back pain (LBP). We hypothesized that IVD mechanical stress and axial loading contribute to discogenic LBP’s pathomechanism. This study aimed to elucidate the relationships among the clinical findings, radiographical findings, and pain-related-molecule expression in human degenerated IVDs. We harvested degenerated-IVD samples from 35 patients during spinal interbody fusion surgery. Pain-related molecules including tumor necrosis factor alpha (TNF-alpha), interleukin (IL)-6, calcitonin gene-related peptide (CGRP), microsomal prostaglandin E synthase-1 (mPGES1), and nerve growth factor (NGF) were determined. We also recorded preoperative clinical findings including body mass index (BMI), Oswestry Disability Index (ODI), and radiographical findings including the vacuum phenomenon (VP) and spinal instability. Furthermore, we compared pain-related-molecule expression between the VP (−) and (+) groups. BMI was significantly correlated with the ODI, CGRP, and mPGES-1 levels. In the VP (+) group, mPGES-1 levels were significantly higher than in the VP (−) group. Additionally, CGRP and mPGES-1 were significantly correlated. Axial loading and mechanical stress correlated with CGRP and mPGES-1 expression and not with inflammatory cytokine or NGF expression. Therefore, axial loading and mechanical stress upregulate CGRP and mPGES-1 in human degenerated IVDs, potentially leading to chronic discogenic LBP.

## 1. Introduction

Low back pain (LBP) is an important clinical issue, and the lifetime prevalence of LBP has been reported to be high and gradually increase as the aging society progresses [1]. In addition, the LBP-induced economic burden has gradually increased, resulting in increased socioeconomic concerns [2,3]. In recent years, 78% of cases were reportedly classified as specific LBP in which the cause is identifiable, and intervertebral discs (IVDs) were considered significant causes of LBP [4].

Regarding the pathomechanism of discogenic LBP, the expression of pain-related molecules is a potential pain-inducing factor. Several studies have shown pain-related molecules including inflammatory cytokines and prostaglandin E2 (PGE2) to be upregulated in herniated or degenerated human IVDs as well as animal models of degenerated IVDs [5,6,7]. Furthermore, nerve growth factor (NGF) and calcitonin gene-related peptide (CGRP) were identified as factors that play an important role in pain development. Newly developed CGRP antagonists and NGF inhibitors have been reported to be effective against pain diseases including headache and musculoskeletal pain [8,9]. Additionally, we previously reported macrophage-derived inflammatory cytokines to regulate the expression of NGF and CGRP in IVD cells, thus potentially influencing LBP [10]. Therefore, various pain-related molecules in IVD cells could be associated with LBP development.

On the other hand, mechanical stress and axial loading on IVDs could also be associated with the development of discogenic LBP. Lumbar segmental instability has clinically been known to be a cause of LBP [11]. Moreover, spinal fusion surgery for lumbar spinal instability has been reported to demonstrate favorable clinical outcome [12]. In addition, we previously reported IVD dynamic compression, which is reflected as mechanical stress and axial loading on IVDs, to lead to long-term upregulation of pain-related molecules including inflammatory cytokines and NGF [13]. Therefore, we sought to determine whether mechanical stress and axial loading on IVDs contribute to the pathomechanism of discogenic LBP.

Based on these findings, we hypothesized that the interaction between the upregulation of pain-related molecules and mechanical stress on IVDs potentially leads to the development of LBP in human IVDs. However, no studies have utilized human samples to evaluate the relationships between the expression of pain-related molecules, mechanical stress, and axial loading on IVDs and LBP. Hence, this study aimed to elucidate the relationships among the clinical findings, radiographical findings, and expression of pain-related molecules in human degenerated IVDs.

## 2. Results

### 2.1. The Relationships among Radiographical Findings, Clinical Findings, and the Expression of Pain-Related Molecules

To evaluate factors associated with mechanical stress and axial loading on IVDs, we focused on body mass index (BMI) and radiographical findings. Table 1 shows the correlation between factors for mechanical stress or axial loading on IVDs and clinical findings or the expression of pain-related molecules. BMI was significantly negatively correlated with the Oswestry Disability Index (ODI) score (r = –0.390, *p* = 0.025). Moreover, BMI was significantly positively correlated with CGRP and microsomal prostaglandin E synthase-1 (mPGES1) (r = 0.383, *p* = 0.028, and r = 0.383, *p* = 0.028, respectively). Furthermore, percentage slippage in the neutral position (%slip) was significantly positively correlated with the visual analogue scale (VAS) of pain in the lower extremity (L/EP), and numbness in the lower extremity (L/EN) (r = 0.412, *p* = 0.019, and r = 0.452, *p* = 0.009, respectively). Nonetheless, there were no significant correlations between %slip and any pain-related molecules (*p* > 0.05). Additionally, translation and slip angles were not correlated with clinical findings or the expression of pain-related molecules (*p* > 0.05).

### 2.2. The Comparison of Clinical Findings and the Expression of Pain-Related Molecules between the Two Groups

Regarding spinal instability, which was defined based on X-ray findings, 19 out of 35 cases had spinal instability. When we divided the cases into instability (+) and (−) groups, pain in the lower extremity (L/EP) visual analogue scale (VAS) and numbness in the lower extremity (L/EN) VAS in the instability (+) group were significantly higher than those in the instability (−) group (*p* = 0.009, 0.017). In addition, the ODI score in the instability (+) group was significantly lower than that in the instability (−) group (*p* = 0.011). However, there were no significant differences in the expression of each of the pain-related molecules between the two groups (*p* > 0.05) (Table 2).

Regarding the presence of vacuum phenomenon (VP), 20 out of 35 cases exhibited VP on computed tomography scans. When we divided cases into VP (+) and (−) groups, there were no significant differences in each of the clinical findings between the two groups (*p* > 0.05). Nevertheless, the expression of mPGES1 in the VP (+) group was significantly higher than that in the VP (−) group (*p* = 0.036), though there were no significant differences in the expression of the other pain-related molecules between the two groups (*p* > 0.05) (Table 3).

### 2.3. Correlations among the Expressions of Various Pain-Related Molecules

The expression of NGF was positively significantly correlated with the expression of TNFA, IL-6, and mPGES1 (r = 0.883, *p* = 0.000; r = 0.424, *p* = 0.028; and r = 0.391, *p* = 0.024, respectively). In addition, the expression of CGRP was positively significantly correlated with the expression of mPGES1 (r = 0.560, *p* = 0.001). However, the expressions of inflammatory cytokines including TNFA and IL-6 were not significantly correlated with the expression of CGRP or mPGES1 (*p* > 0.05) (Table 4).

## 3. Discussion

In the current study, BMI was positively correlated with the expressions of CGRP and mPGES1 in IVDs. Furthermore, mPGES1 expression was higher in IVDs with VP than in those without VP. Additionally, mPGES1 expression was correlated with CGRP and NGF expressions. In contrast, %slip was positively correlated with L/EPVAS and L/ENVAS, but not with the expression of pain-related molecules. Furthermore, cases of radiographical spinal instability exhibited worse clinical findings including L/EPVAS, L/ENVAS, and ODI than those without spinal instability. Moreover, the expression of inflammatory cytokines, including TNFA and IL-6, was positively correlated with the expression of NGF only, but not with any other factors including clinical findings, radiographical findings, and the expressions of CGRP and mPGES1.

Regarding BMI, Webb et al. found high BMI to be one of the most important independent risk factors for LBP and its severity [14]. Hasegawa et al. reported compressive force on IVDs to be an independent risk factor for LBP [15]. According to Nachemson’s report, compressive force on IVDs was dependent on body weight and posture [16]. These findings indicate that excessive axial loading on IVDs potentially induces LBP. In terms of the pathomechanism of axial loading-induced LBP, the present study is the first to evaluate the relationships between axial loading and the expression of pain-related molecules in degenerated IVDs. In the present study, CGRP and mPGES1 expression in degenerated IVDs were correlated with BMI. Furthermore, the expression of mPGES1was positively correlated with that of CGRP in human degenerated IVDs. mPGES1 has been reported to be a terminal enzyme in the biosynthetic pathway of PGE2, which is expressed during inflammation and plays a critical role in diseases associated with pain [17,18]. In addition, CGRP was also reported to be expressed in tissues with musculoskeletal-pain diseases including LBP and osteoarthritis (OA) [7,19]. Additionally, we previously reported PGE2 stimulation to induce the upregulation of CGRP expression, and PGE2 appeared to regulate CGRP receptors in synovial cells in patients with knee OA [20,21]. These findings indicate that axial loading on IVDs induces the upregulation of CGRP and mPGES-1 in human degenerated IVDs, and that PGE2-CGRP signaling may be associated with the pathomechanism of discogenic LBP in humans. Moreover, CGRP, PGE2, and mPGES1 may be potential therapeutic targets for discogenic LBP. Regarding other treatments, Baena-Beato et al. reported that intensive aquatic therapy improved LBP and reduced body weight and BMI [22]. Furthermore, Wasser et al. reported in their review of obese patients with chronic LBP that physical therapy including yoga, Pilates, aerobics, resistance, and aquatic training improved chronic LBP and body composition [23]. Torlak et al. also reported that combining an intermittent diet with physical therapy significantly improved lower back pain and reduced body weight and BMI [24]. These findings indicate that the axial load to the lumbar spine including IVDs decreased after physical therapy and intermittent diet, which may improve LBP.

In terms of VP, VP was defined as the presence of gas in degenerated IVDs, which is associated with progressive IVD degeneration. In a survey targeted at spinal surgeons, Lewandrowski et al. reported that many spinal surgeons considered radiographical VP to be associated with spinal instability [25]. In their clinical study, Liao et al. also concluded that VP should be considered as a sign of spinal instability and treated accordingly [26]. In addition, VP was reported to be a radiographical finding in advanced degenerative disc disease and to be associated with LBP [27]. However, no studies have elucidated the relationships between VP and the expression of pain-related molecules. The current study revealed that mPGES1 in IVDs with VP was upregulated compared with that in IVDs without VP. Based on these findings, combined with previous reports and the current study, VP may be a sign of spinal instability and is potentially associated with the up-regulation of pain-related molecules such as mPGES1.

On the other hand, in the current study, some of the other parameters of spinal instability including %slip, translation, and slip angle were associated with certain clinical findings including ODI score as well as pain and numbness in the L/E, but not with the expression of pain-related molecules in degenerated IVDs. Spinal instability is a well-known, possible risk factor for LBP and sciatica, thus supporting our findings [28,29]. With respect to the relationships between inflammatory cytokines and spinal instability, an association of combined IVD- and anterior longitudinal ligament injury-induced spinal instability with the upregulation of inflammatory cytokines in rabbits has been reported [30]. In addition, we also previously reported IVD injury and dynamic compression to induce the long-lasting upregulation of inflammatory cytokines and NGF in rats [13]. However, we could not detect any pain-related molecules associated with spinal instability in the current study. These discrepancies between previous reports and the current study may be due to differences between animal models and human samples.

In this study, LBP scores did not correlate with pain-related molecules. One reason might be the multifactorial nature of CLBP. Several authors have reported that the causes of LBP in the clinical setting were IVD and myofascial, facet joint, and sacroiliac joint problems [4,31]. Sun et al. reported that IVD degeneration significantly correlates with paraspinal muscle atrophy [32]. Fujiwara et al. also reported that spinal hypermobility causes IVD degeneration and facet joint osteoarthritis [33]. In contrast, some chronic LBP cases have been associated with psychosocial factors [34]. Therefore, in our study, IVD may not be the singular cause of LBP. However, to evaluate the relationships between LBP scores and pain-related molecules in IVDs, confirming the efficacy of discography or discoblock preoperatively was necessary. Another potential reason for our result is IVD innervation. We previously reported an association between the discogenic LBP pathomechanism and innervation, inflammation, and hypermobility [35]. Freemont et al. reported that nerve fibers in the deep layer of IVDs play an important role in the pathogenesis of discogenic chronic LBP [36]. However, we could not evaluate IVD innervation in the current study. IVDs with upregulated pain-related molecules and few sensory nerve fibers might not show severe LBP.

In this study, inflammatory cytokines (e.g., TNFA and IL-6) were not associated with BMI or radiographical factors (e.g., VP and spinal instability). We previously reported that macrophages, potentially originating from bone marrow cells, produce inflammatory cytokines in human degenerated IVDs and mouse injured IVDs [10,37]. Melrose et al. reported a strong association between blood vessel ingrowth into IVDs and proteoglycan depletion [38]. In addition, a previous report showed that inflammatory cytokines including TNFA and IL-6 promote matrix degradation and phenotypic changes in cells, resulting in an imbalance between catabolic and anabolic responses, which leads to IVD degeneration [39]. These findings indicate a close association between macrophage-derived inflammatory cytokine expression and IVD degeneration, contrary to the results of this study. The extracellular matrix (ECM) may explain this discrepancy. Aging and mechanical stress on IVDs have been associated with decreased proteoglycans and collagen [40,41]. In addition, spinal instability induces proteoglycan and collagen depletion, increasing the expression of messenger RNA for collagen, aggrecan, and perlecan [42]. In particular, perlecan plays an important role in ECM stabilization and the mechanoregulation of load-bearing connective tissues [43]. Regarding the relationship between ECM and inflammatory cytokines, several authors have reported associations between inflammatory cytokines expression including TNFA, IL-1B, and IL-17, and mitogen-activated protein kinase (i.e., MAPK) and nuclear factor kappa-light-chain-enhancer of activated B cell (i.e., NF-kappaB) pathway activation, resulting in decreased proteoglycans and collagen in the ECM [44,45]. However, we did not evaluate several ECM molecules including proteoglycans, collagen, aggrecan, and perlecan. Therefore, further studies are warranted.

With regard to the relationships between inflammatory cytokines, NGF, and mPGES1, the expression of NGF was associated with that of TNFA, IL-6, and mPGES1; however, there were no significant correlations between TNFA, IL-6, and mPGES1 in the current study. TNFA stimulation has been reported to induce NGF expression, and NGF promotes the release of inflammatory cytokines [46,47]. Based on these previous reports, the interaction between inflammatory cytokines and NGF may be one of the pathomechanisms underlying pain status. In the current study, axial loading and mechanical stress on IVDs was associated with the expression of mPGES1 and CGRP, and these expressions potentially lead to the expression of inflammatory cytokines via NGF expression (Figure 1).

Axial loading and mechanical stress on intervertebral discs caused by a high body mass index and vacuum phenomenon are associated with the upregulation of calcitonin gene-related peptide (CGRP) and microsomal prostaglandin E synthase-1 (mPGES-1), but not with inflammatory cytokines and nerve growth factor (NGF). mPGES-1 expression correlates with CGRP and NGF expression, and NGF expression correlates with tumor necrosis factor (TNF)-alpha and interleukin-6 (IL-6) expression. These pain-related molecules are potentially associated with the pathomechanism of discogenic low back pain.

This study has certain limitations. First, as above-mentioned, inflammatory-cytokine expression was not associated with radiographical findings, contrary to previous studies that utilized animal models. The small samples in this study might have influenced these discrepancies. Second, in the current study, LBP was evaluated using ODI and VAS scores. As above-mentioned, the pathomechanism underlying LBP may be multifactorial including facet joint-genic and myofascial factors. However, only IVDs were discussed in the current study. In the clinical treatment of LBP, other LBP factors must be considered. Third, although we considered a high BMI and VP as axial loading and mechanical stress, we did not evaluate axial loading in vitro. Fourth, we did not evaluate the levels of pain-related molecules in the blood. Therefore, we could not elucidate the relationship between pain-related molecule expression in the blood and IVDs. Hiyama et al. reported that IL-6 levels in the blood correlated with those in IVDs but not with TNFA [48]. In addition, we only evaluated the mRNA levels of pain-related molecules in IVDs using reverse transcription polymerase chain reaction (RT-PCR); we did not evaluate protein levels using the enzyme-linked immunosorbent assay. Therefore, we could not evaluate differences between the mRNA and protein levels of the pain-related molecules. Further studies are required to confirm this hypothesis. Finally, being the most critical issue, we could not collect control samples of non-degenerated IVDs in the current study. When discussing clinical pain status, normal samples should ideally be used. Further studies using normal samples are therefore warranted.

In conclusion, axial loading and mechanical stress on IVDs due to high BMI and the vacuum phenomenon were associated with the upregulation of CGRP and mPGES-1 but not with inflammatory cytokines and NGF. Furthermore, mPGES-1 expression correlated with CGRP and NGF expression, and NGF expression correlated with TNFA and IL-6 expression. The expression of pain-related factors may lead to discogenic LBP in humans.

## 4. Materials and Methods

### 4.1. Subjects

IVD samples were harvested from 35 patients undergoing spinal interbody fusion surgery or decompression surgery (23 men and 12 women; mean age: 69.9 ± 11.7 years; L1/2: one sample, L2/3: five samples, L3/4: 10 samples, L4/5: 13 samples, and L5/S: six samples) including 19 with lumbar spinal stenosis, 10 with adult spinal deformity, and six with lumbar disc herniation. Twenty-nine IVD samples were harvested from patients with lumbar spinal stenosis and adult spinal deformity during posterior lumbar interbody fusion surgery. Additionally, six IVD samples were harvested from the lumbar disc herniation during decompression surgery. The mean illness duration was 10.2 ± 5.6 months. All patients received more than two types of painkillers including acetaminophen, non-steroidal anti-inflammatory drugs, pregabalin, serotonin noradrenaline reuptake inhibitor, and tramadol. None of the patients had previous spinal surgery.

One of the 35 IVD samples was classified as Pfirrmann Grade 2, one as Pfirrmann Grade 3, 25 as Pfirrmann Grade 4, and eight as Pfirrmann Grade 5 on MRI. All patients complained of sciatica and severe chronic LBP. Six spinal surgeons harvested IVDs using the same procedure. First, the annulus fibrosus was removed using a scalpel and samples were harvested from the center of the IVD.

### 4.2. RT-PCR

Tissue specimens were homogenized using a polytron homogenizer in Trizol solution (Thermo Fisher Scientific, Waltham, MA, USA). After adding chloroform, the homogenized sample was added to a MaXtract™ High Density Tube (Quaigen, Valencia, CA, USA) for phase separation. After centrifugation, there was an aqueous phase from which RNA was obtained by precipitation with isopropanol and a high salt precipitation solution (Takara, Kyoto, Japan). The RNA pellet was washed with 75% ethanol prior to dissolution in diethylpyrocarbonate-treated water. cDNA synthesis was performed using SuperScript™ III RT (Thermo Fisher Scientific). The PCR reactions were performed using TB green Ex Taq (Takara) and primers. The primers were synthesized at Hokkaido System Science Co. (Sapporo, Japan) based on our previous studies [21,49,50,51]. PCR sequences are shown in Table 5 Pain-related molecules including TNF-alpha, IL-6, CGRP, mPGES1, and NGF were determined using RT-PCR. mRNA expression was normalized to the housekeeping gene, glyceraldehyde 3-phosphate dehydrogenase (*GAPDH)*, and mRNA levels using the 2^-ΔΔCT^ method.

### 4.3. Clinical Findings

Preoperative clinical scores were evaluated using the ODI, VAS score of LBP, L/EP, and L/EN. Additionally, we recorded their preoperative body heights and weights in their clinical records and subsequently calculated the BMI.

### 4.4. Radiographical Findings

X-ray images of the neutral–flexion–extension lateral views of the lumbar spine in the lateral position were reviewed to evaluate spinal dynamic instability. To evaluate spinal instability, three parameters including %slip, dynamic translation (which is the difference in segmental slippage between the flexion and extension positions (Translation)), and the slippage angle in the flexion position (Slip angle) were measured as previously reported [52,53] (Figure 2A–C). According to previous reports, spinal instability was defined as >10% of %slip, >3 mm of Translation, or >5° of the Slip angle [52,53,54]. In addition, the presence of the VP on sagittal views of the computed tomography scans was also reviewed (Figure 2D).

### 4.5. Statistical Analysis

We used Pearson’s correlation coefficients to evaluate the relationships between clinical or radiographic findings and pain-related molecule expression and the relationships between each pain-related molecule. The correlation coefficients were classified: r-values of 0.2–0.4, 0.4–0.7, and 0.7-1 were considered weak, moderate, and strong correlations, respectively. Additionally, we divided the patients into groups based on spinal instability (+/–) and VP (+/–) to evaluate the effect of radiographic findings on pain-related molecule expression and clinical findings. We then compared the results between the two groups. Levene’s test was used to assess the variance equality of the variables of interest. The Mann–Whitney U test was used for variables with unequal variances, and an unpaired t-test was used for variables with equal variances. Statistical significance was set at *p* < 0.05.

## Figures and Tables

**Figure 1 ijms-23-02973-f001:**
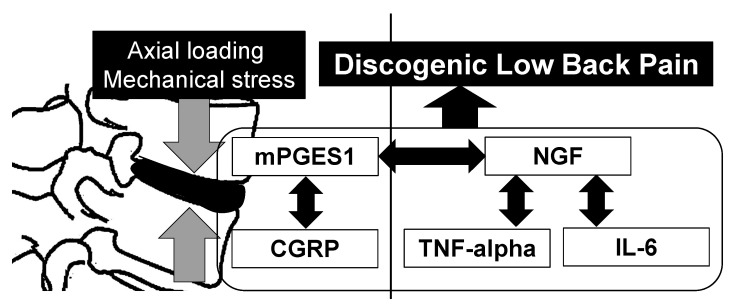
Relationship between mechanical stress on intervertebral discs and pain-related molecule expression.

**Figure 2 ijms-23-02973-f002:**
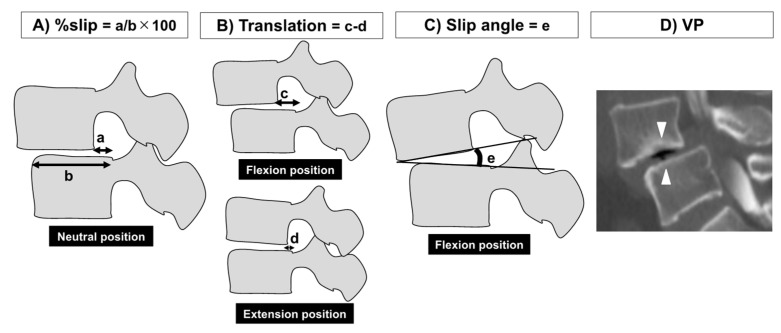
Measurement parameters for evaluating spinal instability. (**A**) Percentage slippage (%slip), (**B**) dynamic translation, which was the difference in segmental slippage between the flexion and extension positions (Translation), (**C**) the slippage angle in the flexion position (Slip angle), and (**D**) the vacuum phenomenon (VP) on sagittal views of computed tomography scans were measured.

**Table 1 ijms-23-02973-t001:** Correlation between factors for mechanical stress or axial loading on IVDs and clinical findings or the expression of pain-related molecules.

		LBPVAS	L/EPVAS	L/ENVAS	ODI	TNFA	IL6	CGRP	NGF	mPGES1
BMI	r	−0.081	0.253	0.102	−0.390	0.328	−0.104	0.383	0.28	0.383
	*p*	0.66	0.163	0.578	0.025	0.063	0.607	0.028	0.115	0.028
%slip	r	0.064	0.412	0.452	−0.261	−0.092	−0.283	−0.204	−0.112	−0.159
	*p*	0.726	0.019	0.009	0.143	0.612	0.152	0.254	0.536	0.376
Translation	r	0.109	0.243	0.078	−0.19	−0.124	−0.011	−0.18	−0.107	−0.053
	*p*	0.552	0.181	0.672	0.29	0.492	0.955	0.317	0.552	0.768
Slip angle	r	−0.108	0.086	−0.144	0.012	0.221	0.247	0.029	0.187	−0.038
	*p*	0.556	0.641	0.433	0.945	0.216	0.215	0.873	0.298	0.832

LBP: low back pain, VAS: visual analogue scale, L/EP: pain in the lower extremity, L/EN: numbness in the lower extremity, ODI: Oswestry Disability Index, TNFA: tumor necrosis factor alpha, IL-6: interleukin-6, CGRP: calcitonin gene-related peptide, NGF: nerve growth factor, mPGES1: microsomal prostaglandin E synthase-1, BMI: body mass index, %slip: the percentage of slippage in the neutral position, Translation: dynamic translation, which was the difference in segmental slippage between the flexion and extension positions, Slip angle: the slippage angle in the flexion position, r: r-value, *p*: *p-*value.

**Table 2 ijms-23-02973-t002:** Comparison of clinical findings and the expression of pain-related molecules between the spinal instability (+) and (−) groups.

	Instability (−) N = 16		Instability (+) N = 19		*p* Value
	Mean	SD	Mean	SD	
LBPVAS	5.400	3.522	6.970	2.265	0.675
L/EPVAS	6.720	3.425	7.320	2.190	0.009
L/ENVAS	5.470	3.079	6.330	3.061	0.017
ODI	57.587	16.994	50.830	12.645	0.011
TNF	0.00140	0.00263	0.00080	0.00116	0.910
IL6	0.00010	0.00010	0.00010	0.00008	0.089
CGRP	0.00080	0.00067	0.00150	0.00278	0.251
NGF	0.03700	0.09173	0.02440	0.05327	0.858
mPGES1	0.01030	0.00687	0.03160	0.04043	0.418

LBP: low back pain, VAS: visual analogue scale, L/EP: pain in the lower extremity, L/EN: numbness in the lower extremity, ODI: Oswestry Disability Index, TNFA: tumor necrosis factor alpha, IL-6: interleukin-6, CGRP: calcitonin gene-related peptide, NGF: nerve growth factor, mPGES1: microsomal prostaglandin E synthase-1.

**Table 3 ijms-23-02973-t003:** Comparison of clinical findings and the expression of pain-related molecules between the vacuum phenomenon (+) and (−) groups.

	VP (−) N = 15		VP (+) N = 20		*p* Value
	Mean	SD	Mean	SD	
LBPVAS	5.4	3.522	6.97	2.265	0.171
L/EPVAS	6.72	3.425	7.32	2.19	0.548
L/ENVAS	5.47	3.079	6.33	3.061	0.444
ODI	57.5873	16.99379	50.8304	12.64459	0.199
TNF	0.0014	0.00263	0.0008	0.00116	0.387
IL6	0.0001	0.0001	0.0001	0.00008	0.486
CGRP	0.0008	0.00067	0.0015	0.00278	0.357
NGF	0.037	0.09173	0.0244	0.05327	0.623
mPGES1	0.0103	0.00687	0.0316	0.04043	0.036

VP: vacuum phenomenon, LBP: low back pain, VAS: visual analogue scale, L/EP: pain in the lower extremity, L/EN: numbness in the lower extremity, ODI: Oswestry Disability Index, TNFA: tumor necrosis factor alpha, IL-6: interleukin-6, CGRP: calcitonin gene-related peptide, NGF: nerve growth factor, mPGES1: microsomal prostaglandin E synthase-1.

**Table 4 ijms-23-02973-t004:** Correlations among the expressions of various pain-related molecules.

		IL6	CGRP	NGF	mPGES1
TNFA	r	0.105	0.065	0.883	0.202
	*p*	0.602	0.718	0.000	0.260
IL6	r		0.113	0.424	−0.021
	*p*		0.573	0.028	0.916
CGRP	r			0.024	0.560
	*p*			0.894	0.001
NGF	r				0.391
	*p*				0.024

TNFA: tumor necrosis factor alpha, IL-6: interleukin-6, CGRP: calcitonin gene-related peptide, NGF: nerve growth factor, mPGES1: microsomal prostaglandin E synthase-1, r: r-value, *p*: *p-*value.

**Table 5 ijms-23-02973-t005:** Sequences of primers used in this study.

Gene	Direction	Primer Sequence (5′–3′)	Product Size (bp)
*TNFA*	F	CTTCTGCCTGCTGCACTTTG	118
	R	GTCACTCGGGGTTCGAGAAG	
*IL6*	F	GAGGAGACTTGCCTGGTGAAA	199
	R	TGGCATTTGTGGTTGGGTCA	
*CGRP*	F	TCCAAAACCCAGAAGACGCA	91
	R	TTGTTCTTCACCACACCCCCTG	
*NGF*	F	CCCATCCCATCTTCCACAGG	74
	R	GGTGGTCTTATCCCCAACCC	
*mPGES1*	F	GGAGACCATCTACCCCTTCCT	81
	R	AAGTGCATCCAGGCGACAAA	
*GAPDH*	F	TGTTGCCATCAATGACCCCTT	202
	R	CTCCACGACGTACTCAGCG	

TNFA: tumor necrosis factor alpha, IL-6: interleukin-6, CGRP: calcitonin gene-related peptide, NGF: nerve growth factor, mPGES1: microsomal prostaglandin E synthase-1.

## Data Availability

The data presented in this study are available on request from the corresponding author.

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
