# Peer review of "A High Body Mass Index and the Vacuum Phenomenon Upregulate Pain-Related Molecules in Human Degenerated Intervertebral Discs"

_ijms, 2022, doi:10.3390/ijms23062973_

Round 1

Reviewer 1 Report

Review of IJMS 1618434

Axial loading and mechanical stress up-regulate pain related molecules in human degenerated intervertebral discs

This is an 18 author review, a large number of authors but a very short paper covering a topical area of investigation and one of considerable importance with discal LBP rightly identified as a major musculoskeletal condition of global socioeconomic significance. This manuscript should be of interest to the IJMS readership and could be improved by discussion of key areas as per my suggestions outlined below. There certainly are sufficient authors at hand to undertake these additions.

Typos/corrections required

Line 103 define ENVAS and EPVAS these are not defined in the Tables. Make sure all other abbreviations are defined at their first point of use in the text.

Statistical significance should be indicated as a capital P in italic.

Table 2 and 5 need to be explained better what do the single letter entries stand for.

The legend to Fig 1 should be better explained, the figure also is poor and should be re-drawn.

The authors should provide a concluding comment or couple of sentences that summarise their major findings to serve as a lasting take-home message to the readership.

More extensive discussion of the impact of their findings in various areas of intervertebral disc pathobiology is required.

The authors should discuss more fully the significance of their findings.

  1. What improvements in rehabilitation are now available?
  2. What about proprioceptive and nociceptive pain?
  3. What about pain generation from paradiscal tissues?
  4. No comments are made on specific effects of IVD instability on disc cell behavior and gene expression.
  5. What cell signaling pathways are involved in the generation of pain molecules?
  6. What alterations in ECM components contribute to pain generating phenomena? Aggrecan, collagen, Sharpeys fibres,versican-fibrillin interactions, elastin, SLRP effects on ECM network sub-structures and regulation of inflammatory cytokine expression?
  7. Perlecan’s roles in biomodulation of intrinsic disc biomechanics and its contributions to IVD stability, cell viability and IVD homeostasis?

Some references that may be useful in these expanded discussion and which the authors may wish to consult are :-

  1. Singh K, Masuda K, Thonar EJ, An HS, Cs-Szabo G. Age-related changes in the extracellular matrix of nucleus pulposus and anulus fibrosus of human intervertebral disc. Spine (Phila Pa 1976). 2009 Jan 1;34(1):10-6.
  2. Taylor TK, Melrose J, Burkhardt D, Ghosh P, Claes LE, Kettler A, Wilke HJ. Spinal biomechanics and aging are major determinants of the proteoglycan metabolism of intervertebral disc cells. Spine (Phila Pa 1976). 2000 Dec 1;25(23):3014-20.
  3. Guilak F, Hayes AJ, Melrose J. Perlecan in Pericellular Mechanosensory Cell-Matrix Communication, Extracellular Matrix Stabilisation and Mechanoregulation of Load-Bearing Connective Tissues. Int J Mol Sci. 2021 Mar 8;22(5):2716.
  4. Melrose J, Shu C, Young C, Ho R, Smith MM, Young AA, Smith SS, Gooden B, Dart A, Podadera J, Appleyard RC, Little CB. Mechanical destabilization induced by controlled annular incision of the intervertebral disc dysregulates metalloproteinase expression and induces disc degeneration. Spine (Phila Pa 1976). 2012 Jan 1;37(1):18-25.
  5. Melrose J, Roberts S, Smith S, Menage J, Ghosh P. Increased nerve and blood vessel ingrowth associated with proteoglycan depletion in an ovine anular lesion model of experimental disc degeneration. Spine (Phila Pa 1976). 2002 Jun 15;27(12):1278-85.
  6. Risbud MV, Shapiro IM. Role of cytokines in intervertebral disc degeneration: pain and disc content. Nat Rev Rheumatol. 2014 Jan;10(1):44-56.

Author Response

Responses to the reviewers’ comments:

Reviewer #1

This is an 18 author review, a large number of authors but a very short paper covering a topical area of investigation and one of considerable importance with discal LBP rightly identified as a major musculoskeletal condition of global socioeconomic significance. This manuscript should be of interest to the IJMS readership and could be improved by discussion of key areas as per my suggestions outlined below. There certainly are sufficient authors at hand to undertake these additions.

Response.

We thank the reviewer for taking the time to evaluate our manuscript, which we have revised following your suggestions. Specifically, we have expanded the Discussion section. Below, we provide point-by-point responses to these concerns. The underlined text has been taken directly from the revised manuscript for your review.

Line 103 define ENVAS and EPVAS these are not defined in the Tables. Make sure all other abbreviations are defined at their first point of use in the text.

Response.

We have defined the abbreviations mentioned by the reviewer and confirm that all other abbreviations are defined at their first point of use.

Lines 104-105,

pain in the lower extremity (L/EP) visual analogue scale (VAS) and numbness in the lower extremity (L/EN) VAS

Statistical significance should be indicated as a capital P in italic.

Response.

We have revised the formatting of all P-values in the text.

Table 2 and 5 need to be explained better what do the single letter entries stand for.

Response.

We have added the following explanation regarding the single letter entries in Tables 2 and 5:

r: r-value, P: P-value

The legend to Fig 1 should be better explained, the figure also is poor and should be re-drawn.

Response.

As per the reviewer’s suggestion, we have redrawn Figure 1 and revised the legend.

Lines 257-265,

Figure 1. Relationship between mechanical stress on intervertebral discs and pain-related-molecule expression.

Axial loading and mechanical stress on intervertebral discs caused by a high body mass index and vacuum phenomenon are associated with the upregulation of calcitonin gene-related peptide (CGRP) and microsomal prostaglandin E synthase-1 (mPGES-1), but not with inflammatory cytokines and nerve growth factor (NGF). mPGES-1 expression correlates with CGRP and NGF expression, and NGF expression correlates with tumor necrosis factor (TNF)-alpha and interleukin-6 (IL-6) expression. These pain-related molecules are potentially associated with the pathomechanism of discogenic low back pain.

The authors should provide a concluding comment or couple of sentences that summarise their major findings to serve as a lasting take-home message to the readership.

Response.

As per the reviewer’s suggestion, we have revised the conclusion sentence to emphasize our take-home message.

Lines 286-290,

In conclusion, axial loading and mechanical stress on IVDs due to high BMI and the vacuum phenomenon were associated with the upregulation of CGRP and mPGES-1 but not inflammatory cytokines and NGF. Furthermore, mPGES-1 expression correlated with CGRP and NGF expression, and NGF expression correlated with TNFA and IL-6 expression. The expression of pain-related factors may lead to discogenic LBP in humans.

More extensive discussion of the impact of their findings in various areas of intervertebral disc pathobiology is required. The authors should discuss more fully the significance of their findings.

What improvements in rehabilitation are now available?

Response.

Based on our study results, we also propose that rehabilitation for high body weight and BMI could benefit CLBP. Therefore, we have expanded on this idea in the Discussion section and included additional references.

Lines 173-181,

Regarding other treatments, Baena-Beato et al. reported that intensive aquatic therapy improved LBP and reduced body weight and BMI. Furthermore, Wasser et al. reported in their review of obese patients with chronic LBP that physical therapy, including yoga, Pilates, and aerobic, resistance, and aquatic training, improved chronic LBP and body composition. Torlak et al. also reported that combining an intermittent diet with physical therapy significantly improved lower back pain and reduced body weight and BMI These findings indicate that the axial load to the lumbar spine, including IVDs, decreased after physical therapy and intermittent diet, which may improve LBP.

What about proprioceptive and nociceptive pain?

What about pain generation from paradiscal tissues?

Response.

We have expanded the Discussion section and included new references to address these points.

Lines 208-221,

In this study, LBP scores did not correlate with pain-related molecules. One reason might be the multifactorial nature of CLBP. Several authors have reported that the causes of LBP in the clinical setting were IVD and myofascial, facet joint, and sacroiliac joint problems. Sun et al. reported that IVD degeneration significantly correlates with paraspinal muscle atrophy. Fujiwara et al. also reported that spinal hypermobility causes IVD degeneration and facet joint osteoarthritis. Contrastingly, some chronic LBP cases have been associated with psychosocial factors. Therefore, in our study, IVD may not be the singular cause of LBP. However, to evaluate the relationships between LBP scores and pain-related molecules in IVDs, confirming the efficacy of discography or discoblock preoperatively was necessary. Another potential reason for our result is IVD innervation. We previously reported an association between the discogenic LBP pathomechanism and innervation, inflammation, and hypermobility. Freemont et al. reported that nerve fibers in the deep layer of IVDs play an important role in the pathogenesis of discogenic chronic LBP. However, we could not evaluate IVD innervation in the current study. IVDs with upregulated pain-related molecules and few sensory nerve fibers might not show severe LBP.

No comments are made on specific effects of IVD instability on disc cell behavior and gene expression.

What cell signaling pathways are involved in the generation of pain molecules?

What alterations in ECM components contribute to pain generating phenomena? Aggrecan, collagen, Sharpeys fibres,versican-fibrillin interactions, elastin, SLRP effects on ECM network sub-structures and regulation of inflammatory cytokine expression?

Perlecan’s roles in biomodulation of intrinsic disc biomechanics and its contributions to IVD stability, cell viability and IVD homeostasis?

Response.

As per the reviewer’s suggestion, we have expanded the discussion section and included new references to address these topics.

Lines 222-243,

In this study, inflammatory cytokines (e.g., TNFA and IL-6) were not associated with BMI or radiographical factors (e.g., VP and spinal instability). We previously reported that macrophages, potentially originating from bone marrow cells, produce inflammatory cytokines in human degenerated IVDs and mouse injured IVDs. Melrose et al. reported a strong association between blood vessel ingrowth into IVDs and proteoglycan depletion. In addition, a previous report showed that inflammatory cytokines, including TNFA and IL-6, promote matrix degradation and phenotypic changes in cells, resulting in an imbalance between catabolic and anabolic responses, which leads to IVD degeneration. These findings indicate a close association between macrophage-derived inflammatory cytokine expression and IVD degeneration, contrary to the results of this study.

              The extracellular matrix (ECM) may explain this discrepancy. Aging and mechanical stress on IVDs have been associated with decreased proteoglycans and collagen. In addition, spinal instability induces proteoglycan and collagen depletion, increasing the expression of messenger RNA for collagen, aggrecan, and perlecan. In particular, perlecan plays an important role in ECM stabilization and the mechanoregulation of load-bearing connective tissues. Regarding the relationship between ECM and inflammatory cytokines, several authors have reported associations between inflammatory cytokines expression, including TNFA, IL-1B, and IL-17, and mitogen-activated protein kinase (i.e., MAPK) and nuclear factor kappa-light-chain-enhancer of activated B cells (i.e., NF-kappaB) pathway activation, resulting in decreased proteoglycans and collagen in the ECM. However, we did not evaluate several ECM molecules, including proteoglycans, collagen, aggrecan, and perlecan. Therefore, further studies are warranted.

If one of the referees has suggested that your manuscript should undergo extensive English revisions, please address this issue during revision. We propose that you use one of the editing services listed at https://www.mdpi.com/authors/english or have your manuscript checked by a native English-speaking colleague.

Response.

Our manuscript, including the revised version, has been edited by a native English speaker, and we have attached the certificate files.

Reviewer 2 Report

Authors present an in-vitro study on degenerated samples od intervertebral disc (IVD) from 35 patients during spinal interbody fusion surgery where pain-related molecules, including tumor necrosis factor alpha (TNF-alpha), interleukin (IL)-6, calcitonin gene-related peptide (CGRP), microsomal prostaglandin E synthase-1 (mPGES1), and nerve growth factor (NGF), were determined and correlated to preoperative clinical findings, including body mass index (BMI), Oswestry Disability Index (ODI), and radiographical findings, including the vacuum phenomenon (VP) and spinal instability. Hypothesis was that axial loading and mechanical stress upregulate these molecules in IVDs with effect on mechanism on low back pain (LBP).
BMI was significantly correlated with ODI, CGRP, and mPGES-1 levels. In the VP (+) group, mPGES-1 levels were significantly higher than in the VP (–) group. CGRP and mPGES-1 were significantly correlated. Axial loading and mechanical stress were found to correlate with CGRP and mPGES-1 
expressions . 

Important drawback of this study is low number of patients, so I would change the title into Axial loading and mechanical stress MAY UPREGULATE. However, one should also consider the title Axial loading and mechanical stress to change into Spinal instability - because axial loading and mechanical stress are not in-vitro examined, but the authors claim these patients had axial loading and stress due to instability - OR was there any kind of preoperative test or in-vitro test of axial loading the authors did not report or explained fully?

Further important issue is that Materials and Methods part is on the wrong place in the manuscript (at the end), please insert like usual before Results. Please add thorough explanation on type of assessment of pain related molecules (tests, technique) and were these pain-related molecules in the IVD itself and why was not there a correlation to levels of these molecules in blood, which is crucial. For 35 patients, age, gender and type of fusion surgery should be added - it is also relevant if these patients were chronic pain patients, how long did they take analgesia prior to surgery and most important thing - did they had previous surgeries on the spine and did they have previous conditions which could contribute to upregulation of pain molecules? 

Furthermore, please explain thoroughly the way the correlations and comparisons were made - which statistical tests and which methods, two sentences are not sufficient. 

Author Response

Responses to the reviewers’ comments:

Reviewer #2

Authors present an in-vitro study on degenerated samples of intervertebral disc (IVD) from 35 patients during spinal interbody fusion surgery where pain-related molecules, including tumor necrosis factor alpha (TNF-alpha), interleukin (IL)-6, calcitonin gene-related peptide (CGRP), microsomal prostaglandin E synthase-1 (mPGES1), and nerve growth factor (NGF), were determined and correlated to preoperative clinical findings, including body mass index (BMI), Oswestry Disability Index (ODI), and radiographical findings, including the vacuum phenomenon (VP) and spinal instability. Hypothesis was that axial loading and mechanical stress upregulate these molecules in IVDs with effect on mechanism on low back pain (LBP).

BMI was significantly correlated with ODI, CGRP, and mPGES-1 levels. In the VP (+) group, mPGES-1 levels were significantly higher than in the VP (–) group. CGRP and mPGES-1 were significantly correlated. Axial loading and mechanical stress were found to correlate with CGRP and mPGES-1 expressions.

Response.

We thank the reviewer for taking time to evaluate our manuscript, which we have revised following your suggestions. Please see below for our point-by-point responses.

Important drawback of this study is low number of patients, so I would change the title into Axial loading and mechanical stress MAY UPREGULATE. However, one should also consider the title Axial loading and mechanical stress to change into Spinal instability - because axial loading and mechanical stress are not in-vitro examined, but the authors claim these patients had axial loading and stress due to instability - OR was there any kind of preoperative test or in-vitro test of axial loading the authors did not report or explained fully?

Response.

We agree with your suggestion and have revised the title of the paper.

Title: A high body mass index and the vacuum phenomenon upregulate pain-related molecules in human degenerated intervertebral discs

Moreover, we did not evaluate axial loading in vitro and consider this a study limitation. Therefore, we have addressed this as a limitation in the Discussion section.

Lines 272-273,

Third, although we considered a high BMI and VP as axial loading and mechanical stress, we did not evaluate axial loading in vitro.

Further important issue is that Materials and Methods part is on the wrong place in the manuscript (at the end), please insert like usual before Results.

Response.

The International Journal of Molecular Science template indicates that the Materials and Methods section should be placed after the Results and Discussion section. Therefore, we retained this format.

Please add thorough explanation on type of assessment of pain related molecules (tests, technique) and were these pain-related molecules in the IVD itself and why was not there a correlation to levels of these molecules in blood, which is crucial.

Response.

In the current study, pain-related molecule expression in IVDs was determined by RT-PCR but not by ELISA. In addition, we did not evaluate the levels of pain-related molecules in the blood. Therefore, we could not elucidate the relationship between pain-related molecule expression in the blood and IVDs, which was a study limitation. Thus, we addressed this in more detail in the Discussion section and included additional references.

Fourth, we did not evaluate the levels of pain-related molecules in the blood. Therefore, we could not elucidate the relationship between pain-related molecule expression in the blood and IVDs. Hiyama et al. reported that IL-6 levels in the blood correlated with those in IVDs but not with TNFA. In addition, we only evaluated the mRNA levels of pain-related molecules in IVDs using reverse transcription polymerase chain reaction (RT-PCR); we did not evaluate protein levels using enzyme-linked immunosorbent assay. Therefore, we could not evaluate differences between mRNA and protein levels of the pain-related molecules. Further studies are required to confirm this hypothesis.

For 35 patients, age, gender and type of fusion surgery should be added - it is also relevant if these patients were chronic pain patients, how long did they take analgesia prior to surgery and most important thing - did they had previous surgeries on the spine and did they have previous conditions which could contribute to upregulation of pain molecules?

Response.

We agree that this is important information. Therefore, we detailed the patient demographics in the Materials and Methods section.

Lines 293-303,

IVD samples were harvested from 35 patients undergoing spinal interbody fusion surgery or decompression surgery (23 men and 12 women; mean age: 69.9±11.7 years; L1/2: 1 sample, L2/3: 5 samples, L3/4: 10 samples, L4/5: 13 samples, and L5/S: 6 samples), including 19 with lumbar spinal stenosis, 10 with adult spinal deformity, and 6 with lumbar disc herniation. Twenty-nine IVD samples were harvested from patients with lumbar spinal stenosis and adult spinal deformity during posterior lumbar interbody fusion surgery. Additionally, six IVD samples were harvested from the lumbar disc herniation during decompression surgery. The mean illness duration was 10.2 ± 5.6 months. All patients received more than two types of painkillers, including acetaminophen, non-steroidal anti-inflammatory drugs, pregabalin, serotonin noradrenaline reuptake inhibitor, and tramadol. None of the patients had previous spinal surgery.

Furthermore, please explain thoroughly the way the correlations and comparisons were made - which statistical tests and which methods, two sentences are not sufficient.

Response.

We have revised the statistical section to include more detail.

We used Pearson’s correlation coefficients to evaluate the relationships between clinical or radiographic findings and pain-related molecule expression and the relationships between each pain-related molecule. The correlation coefficients were classified based on the report by Dancey et al.; r-values of 0.1–0.3, 0.3–0.6, 0.6–0.9, and 1 were considered weak, moderate, strong, and perfect correlations, respectively. Additionally, we divided the patients into groups based on spinal instability (+/–) and VP (+/–) to evaluate the effect of radiographic findings on pain-related molecule expression and clinical findings. We then compared the results between the two groups. Leven’s test was used to assess the variance equality of the variables of interest. The Mann–Whitney U test was used for variables with unequal variances, and an unpaired t-test was used for variables with equal variances. Statistical significance was set at P <0.05.

If one of the referees has suggested that your manuscript should undergo extensive English revisions, please address this issue during revision. We propose that you use one of the editing services listed at https://www.mdpi.com/authors/english or have your manuscript checked by a native English-speaking colleague.

Response.

Our manuscript, including the revised version, has been edited by a native English speaker, and we have attached the certificate files.

Round 2

Reviewer 1 Report

The authors have responded to all of the points I raised and in my opinion it is considerably improved.